# Compact Dual-Band Antenna with Paired L-Shape Slots for On- and Off-Body Wireless Communication

**DOI:** 10.3390/s21237953

**Published:** 2021-11-29

**Authors:** Sarosh Ahmad, Adnan Ghaffar, Niamat Hussain, Nam Kim

**Affiliations:** 1Department of Signal Theory and Communications, Universidad Carlos III de Madrid (UC3M), 28903 Madrid, Spain; saroshahmad@ieee.org; 2Department of Electrical Engineering and Technology, Government College University Faisalabad (GCUF), Faisalabad 38000, Pakistan; 3Department of Electrical and Electronic Engineering, Auckland University of Technology, Auckland 1010, New Zealand; aghaffar@aut.ac.nz; 4Department of Information and Communication Engineering, Chungbuk National University, Cheongju 28644, Korea; hussain@chungbuk.ac.kr

**Keywords:** ISM band, dual-band, on/off-body, dual-mode, wireless networks

## Abstract

A simple dual-band patch antenna with paired L-shap slots for on- and off-body communications has been presented in this article. The proposed antenna resonates in the industrial, scientific, and medical (ISM) band at two different frequencies, at 2.45 GHz and 5.8 GHz. At the lower frequency band, the antenna’s radiation pattern is broadsided directional, whereas it is omni-directional at the higher frequency band. The efficiency and performance of the proposed antenna under the influence of the physical body are improved, and the specific absorption rate (SAR) value is significantly reduced by creating a full ground plane behind the substrate. The substrate’s material is FR-4, the thickness of which is 1.6 mm and it has a loss tangent of tanδ = 0.02. The overall size of the proposed design is 40 mm × 30 mm × 1.6 mm. Physical phantoms, such as skin, fat and muscle, are used to evaluate the impact of physical layers at 2.45 GHz and 5.8 GHz. The SAR values are assessed and found to be 0.19 W/kg and 1.18 W/kg at 2.45 GHz and 5.8 GHz, respectively, over 1 gram of mass tissue. The acquired results indicate that this antenna can be used for future on- and off-body communications and wireless services.

## 1. Introduction

Body-centric communication (BCC) may be separated into three different areas according to the means of communication, specifically, in-, off- and on-body communication, as studied in [1,2,3]. Antennas and sensors are vital components of BCC systems. At first, sensors were embedded into physical body phantoms, and the related proposed antenna was an implantable gadget with a recidivist on the human skin. The other form is an on-body correspondence that refers to a circumstance where various sensors are kept on the body, and it requires self-communication within the body area network (BAN) [4,5,6]. The third method is the most useful, transmitting information from sensors (antennas) into the atompshere about the physical frame, for example, the monitoring center or repair appliance, and communicating to introduce a personal area network among various people and adjacent devices [7,8,9,10]. Fundamentally, the efficiency and performance of the designed antenna under the influence of the physical body are minimized, and the SAR’s value is significantly reduced. Nowadays, off-body antenna for medical applications play a very critical role in improving the living abilities of patients. For radiocommunication between biomedical implementations and exterior display gadgets, a telecommunication scheme using a far-field antenna offers various benefits over its conservative near-field counterpart, such as long-distance communication and high data rates [11,12,13,14,15,16]. A dual-layer substrate-based dual resonating antenna for on- and off-body communications was presented in [17]. The operating frequencies of the antenna were 2.45 and 5.8 GHz, with operating bandwidths of 4.2% and 10.2%, respectively. The material used for the substrate was FR-4 with a 0.6 mm-thickness. The overall size of the antennas was 50 × 50 × 3.2 mm^3^. In [18], a dual-band circular patch antenna for on- and off-body wireless networks was explained. The substrate material used was felt, with a thickness of 2 mm, and the overall volume was 100 × 100 × 2 mm^3^. The radiating patch was a simple slotted circular patch that operated at 2.45 and 5.8 GHz. However, the volume of the proposed antenna was very large. In [19], a body-worn belt for on-body utilization was reported. This belt antenna functioned at 2.45 GHz. However, the antenna produced back radiation that was harmful to the human body, so an electromagnetic bandgap (EBG) technique was utilized to decrease the back radiation of the antenna. The addition of the EBG increased the overall size of the antenna. In [20], a D-shaped slotted patch antenna for on- and off-body communications was proposed. The antenna resonated at the 2.45 and 5.8 GHz frequency bands. The antenna had an omni-directional pattern in the lower band and a directional radiation pattern n the higher band. A dual resonating antenna for application in a wireless body area network (WBAN) was stated in [21]. The antenna consisted of dual layer FR-4 substrate connected via shorting pins. There was a circular patch on the first layer of the substrate, while a quadrilateral patch was placed on the second layer of the substrate. Three shorting pins were used to connect both layers of the substrates. The antenna worked at 2.45 GHz and 5.8 GHz, with an overall size of 30 × 45 × 3.2 mm^3^. Another felt substrate-based dual-band antenna operational at 2 and 5.8 GHz for wearable utilization was presented in [22]. The antenna achieved good gain and bandwidth, but the antenna was two times bigger than our proposed design. The overall volume of the antenna was 80 × 92 × 2 mm^3^, but the SAR values were not evaluated in this study. An F4B substrate-based antenna of 100 × 100 × 3.2 mm^3^ was depicted in [23]. This dual-band antenna operated at 2.45 and 5.8 GHz, with impedance bandwidths of 2.57% and 5.22%, respectively, and the SAR values were not analyzed. The antenna consisted of a slotted circular patch with split-ring resonators (SRR), and there was a circular slot inside the ground plane. In [24], a textile-based dual-band antenna based on SRR for WBAN applications was presented. There was a G-shaped slot inside the radiating patch, and it was printed on a felt substrate with a height of 3 mm. The antenna operated at 2.45 and 3.5 GHz, with bandwidths of 5.3% and 3.14%. 

In this article, an on- and off-body communication dual-mode, dual-band antenna is proposed. The basic principle of the antenna is to utilize two specific modes (on, and off) for the operation of the rectangular patch antenna at two different frequency bands. The radiating patch and the ground plane are on opposite sides of the substrate. The stripline feed technique is used, which is enabled by the SMA connector. L-shaped slots are used inside the patch to resonate the proposed antenna at 2.45 GHz and 5.8 GHz, respectively. The designed antenna is made up of three distinct layers, namely, a ground-plane, a patch, and substrates. At a lower frequency band, the antenna’s radiation pattern is broadsided directional, whereas it is omni-directional at the higher frequency band. The results indicate that this antenna can be utilized for applications in WBAN. This antenna is miniaturized and assembled for use with on- and off-body links. The analysis of the design of the dual-band antenna is outlined in Section 2. Antenna testing under human proximity is illustrated in Section 3, while the conclusion is presented in Section 4. 

## 2. Antenna Design Analysis

A schematic diagram of the designed patch antenna is given in Figure 1. The simulations and optimizations have been supported by computer simulation technology (CST) software. At first, a simple square patch was designed, then L-shaped slots were introduced inside the top and bottom sides of the patch to resonate the proposed antenna at 2.45 GHz and 5.8 GHz. The antenna was fabricated on a lower-priced substrate called FR-4 (relative permittivity of 4.3, loss tangent of 0.025) of 1.6 mm thickness. The size of the designed antenna was 40 × 30 × 1.6 mm^3^. The proposed antenna’s parameters are listed in Table 1.

The antenna’s reflection coefficient is given in Figure 2, and it can be observed that it resonates at 2.45 GHz and 5.8 GHz. The reflection coefficients are −33.2 dB at 2.45 GHz and −24.8 dB at 5.8 GHz. The operational bandwidth of the designed antenna is from 2.4 GHz to 2.48 GHz (3.265%) at 2.45 GHz, and from 5.72 GHz to 5.9 GHz (3.12%) at 5.8 GHz. The *S*_11_ for ideal antennas should be negative infinity [2]. The design process of the patch antenna is explained as follows: 

The primary antenna design (ANT I) shown in Figure 3a contains a 50-Ω simple feedline, a rectangular patch, and the ground plane. 

The patch’s width and length are calculated using Equations (1) and (2) [23], as follows: (1)$$Wp=\frac{\lambda_{o}}{2(\sqrt{0.5\left(\varepsilon_{r}+1\right)})}$$
where $\varepsilon_{r}$ and $\lambda_{o}$ are the relative permittivity and the wavelength of the substrate in free space at the operating frequency. The best choice of *Wp* enables perfect impedance matching. The length of the patch can be evaluated using Equation (2).
(2)$$Lp=\frac{c_{o}}{2f_{o}\sqrt{\varepsilon_{eff}}}-2\Delta L_{p}$$
where $c_{o}$, $\Delta L_{p}$, and $\varepsilon_{eff}$ are the velocity of light, the change in the length of the patch due to its fringing effect, and the effective dielectric constant, respectively. The effective relative permittivity can be calculated using Equation (3).
(3)$$\varepsilon_{eff}=\frac{\varepsilon_{r}+1}{2}+\frac{\varepsilon_{r}-1}{2}\left(\frac{1}{\sqrt{1+12\frac{hs}{W_{p}}}}\right)$$
where *hs* is the height of the substrate. In the end, the fringing effect can be calculated using Equation (4)
(4)$$\Delta Lp=0.421h_{s}\frac{(\varepsilon_{eff}+0.300)\left(\frac{W_{p}}{hs}+0.264\right)}{(\varepsilon_{eff}-0.258)\left(\frac{W_{p}}{hs}+0.813\right)}$$
with the assignment of $\varepsilon_{r}=4.3$ and *h_s_* = 1.6 mm in (1)–(4), the initial parameters of the rectangular patch are *L_p_* = 23 mm mm and *W_p_* = 24 mm. With a simple rectangular patch, the antenna works only at 6 GHz with minimum return loss, as illustrated in Figure 4. As such, to generate two bands, it is necessary to create slots inside the radiating element of the antenna. Now, in the second step (ANT II), L-shaped slots are introduced in the upper side of the radiating patch to shift the lower frequency range from 3 GHz to 2.55 GHz (450 MHz). Then, in the third step (ANT III), L-shaped slots are introduced in the lower side of the radiating patch to operate the antenna at two operating bands, i.e., 2.45 GHz and 5.8 GHz. 

### 2.1. Parametric Study of the Proposed Antenna

The parametric optimization of the antenna is presented step by step. The antenna is simulated by changing the values of the lower horizontal slot *“Lhs”*, upper horizontal slot *“Uhs”*, length of the patch *“L1”*, and the width of the patch *“W1”*, as can be seen in Figure 5. The width of the upper horizontal slot *“Uhs”* is changed from 2.2 to 4.2 mm, then the upper frequency band is kept the same while the lower frequency band is shifted from 2.3 GHz to 2.6 GHz (300 MHz). As can be observed in Figure 5a, the optimum value is 3.2 mm. Similarly, when the width of the lower horizontal slot *“Lhs”* is changed from 1 to 3 mm, the upper frequency band remains the same while the lower frequency band shifts from 2.35 GHz to 2.65 GHz (300 MHz); as can be seen in Figure 5b, the optimum value is 2 mm. The length of the patch *“L1”* is changed from 22 to 24 mm, and we noticed that by increasing the value of the patch length, the upper frequency band shifts from 5.7 GHz to 5.9 GHz, but the lower frequency band remains the same, as can be seen in Figure 5c. In the other case, when the patch’s width *“W1”* is varied from 11 to 13 mm, the lower frequency band shifts from 2.34 GHz to 2.5 GHz (160 MHz), as depicted in Figure 5d. In the last case, when the vertical slot *“L2”* is varied from 8 to 10 mm, both frequency bands slightly shift by about 100 MHz, as depicted in Figure 5e.

The surface current density of an antenna indicates which section of the antenna is playing the most significant role in making it resonate at the desired frequency. The red coloration shows that that portion of the antenna is making a large contribution. Therefore, if you make changes in that portion, then your frequency will vary. For example, the surface current density at 2.45 GHz is illustrated in Figure 6a. The upper portion of the patch and the feedline play a significant role in making the antenna resonate at 2.45 GHz. Similarly, at 5.8 GHz, the lower portion of the slotted patch plays a major part in making this antenna resonate at the desired frequency (see Figure 6b).

### 2.2. Equivalent Circuit Model

A circuit model for the proposed dual-band antenna for on- and off-body communications is presented in Figure 7. The circuit model is designed using an advanced design system (ADS) software. The circuit model consists of four inductors, three capacitors, three resistors, and two resistor–inductor–capacitor (RLC) circuits connected in parallel with each other, as given in Figure 7a. By varying the values of the resistors, the *S*_11_ of the circuit model can be varied, while by changing the values of the capacitors and inductors, the *S*_11_ of the antenna can be tuned. The left-sided circuit is designed for the lower frequency band, and consists of an RLC circuit connected with one inductor and a capacitor, operating at 2.45 GHz. Similarly, the right-sided circuit is responsible for the higher frequency band (5.8 GHz), which is also an RLC circuit connected with an inductor and a resistor. The reflection coefficient of the circuit model and the antenna’s *S*_11_ are in close agreement with each other, as can be seen in Figure 7b. It covers the bandwidth from 2.41 GHz to 2.47 GHz (60 MHz) at 2.45 GHz, and 5.73 GHz to 5.88 GHz (150 MHz) at 5.8 GHz. The values of the lumped elements are listed in Table 2.

#### Parametric Analysis of the Circuit Model

The optimization of the circuit components and their impacts on the resonance of the antenna are explained in this section. Firstly, in order to vary the lower frequency band (2.45 GHz), we optimized the values of the components *“L2”*, *“C2”*, *“R2”*, *“Lin”*, and *“Cin”*. When the value of *“L2”* is reduced from 2 nH to 1 nH, the lower frequency band shifts from 2.45 GHz to 2.15 GHz, while the higher frequency band remains the same. When the values of *“C2”*, *“R2”*, *“Lin”*, and *“Cin”* are reduced, the lower frequency band moves towers the upper frequency band, as can be seen in Figure 8a. A similar phenomenon can be observed in the parametric study of the physical parameters of the antenna (Figure 5a,b,d). There is a shift in the lower band alone when the values of the upper horizontal slot *“UHS”*, lower horizontal slot *“LHS”*, and width *“W1”* are changed. That means that these parameters (*“UHS*, *LHS*, and *W1”*) correspond to *“C2”*, *“R2”*, *“Lin”*, and *“Cin”* in the equivalent circuit. In the case of the higher frequency band (5.8 GHz), when the value of *“Lout2* is decreased from 3 nH to 2 nH, there is a shift in the higher frequency band from 5.8 GHz to 5.98 GHz (180 MHz), as seen in Figure 8b. By varying the values of *“L1*, *C1*, *R1*, and *Rout”*, the higher frequency band can be tuned. The same thing happens when the values of the lower horizontal slot *“LHS”*, length of the patch *“L1”*, and length of the slot *“L2”* are varied Figure 5b,c,e. This implies that the values of *“L1*, *C1*, *R1*, and *Rout”* depend on the physical dimensions of *“L1*, *C1*, *R1*, and *Rout”.*

## 3. Fabrication and Measurements

### 3.1. Antenna Testing in Free Space

A photograph of the fabricated prototype based on the FR-4 substrate is presented in Figure 9, and the experimental *S*_11_ is shown in Figure 10. From the figures, it can be infered that the measured |*S*_11_| (dB) is close to the simulated |*S*_11_| (dB). In the case of the simulation results in free space, the presented antenna covers the bandwidths from 2.4 GHz to 2.48 GHz (3.265%) at 2.45 GHz, and from 5.72 GHz to 5.9 GHz (3.12%) at 5.8 GHz, while in the case of measurement results, the antenna covers the bandwidths 2.42–2.483 GHz (2.57%) at 2.45 GHz and 5.75–5.9 GHz (2.58%) at 5.8 GHz, as illustrated in Figure 10. 

The *E*- and *H*-planes of the antenna in free space are simulated and measured in Figure 11. The patterns show an omni-directional arrangement (along the *E*-plane) and a directional arrangement (along *H*-plane) at the lower frequency band (2.45 GHz). On the other hand, a broadsided directional pattern emerges along the *E*-plane and an isotropic pattern along the *H*-plane at 5.8 GHz. The red-colored (dotted) line indicates the measured radiation pattern, and the blue colored (solid) line shows the simulated radiation pattern in the *E*- and *H*-planes. The simulated peak gains of the antenna are found to be 5.08 dBi at 2.45 GHz and 6.32 dBi at 5.8 GHz, while the measured peak gains are calculated to be 4.99 dBi and 6.25 dBi at 2.45 GHz and 5.8 GHz, respectively. 

### 3.2. Antenna Testing under Human Proximity

In this section, on- and off-body antenna testing are studied. The area of the skin, fat, and muscle are kept the same, at 100 × 100 mm^2^. The height of the tissues, i.e., skin, fat, and muscle, are kept to 2 mm, 3 mm and 8 mm, correspondingly. The antenna is simulated and tested on the skin, and the dielectric constant of the skin is kept at 41.4 with thermal conductivity of 0.88 S/m, as shown in Figure 12a. After testing the antenna on the skin, we noticed that the gain of the antenna shifts from 5.08 dBi to 3.15dBi and from 6.33 dBi to 7.52 dBi at 2.45 GHz and 5.8GHz, respectively. In the simulations on the human body, the presented antenna covers the bandwidths of 2.42 GHz to 2.47 GHz (2.04%) at 2.45 GHz and from 5.76 GHz to 5.96 GHz (3.44%) at 5.8 GHz, while in the case of the measurement results, the antenna covers the bandwidths 2.44 GHz to 2.473 GHz (1.35%) at 2.45 GHz and from 5.76 GHz to 5.85 GHz (1.55%) at 5.8 GHz, as illustrated in Figure 12b.

The *E*- and *H*-planes of the antenna on human tissue are simulated and measured in Figure 13. The patterns show a broadsided directional arrangement at the lower frequency band (2.45 GHz) and an omni-directional radiation arrangement at the higher frequency band (5.8 GHz). The red-colored (dotted) line indicates the measured radiation pattern, and the blue-colored (solid) line shows the simulated radiation pattern, for both the *E*- and *H*-planes. The simulated peak gains of the antenna are found to be 3.15 dBi at 2.45 GHz and 7.52 dBi at 5.8 GHz, while the measured peak gains are calculated to be 3.08 dBi and 7.34 dBi at 2.45 GHz and 5.8 GHz, correspondingly. 

### 3.3. SAR Analysis

The specific absorption rate is a quantification of how much radiofrequency energy is absorbed by human tissue when it is delivered. It is determined by taking an average over a given volume of 1 gram or 10 grams. The SAR limit in the United States is 1.6 W/kg for 1 gram of tissue, while in Europe, it is 2 W/kg for 10 grams of tissue. The SAR values can be evaluated using Equation (5) [24]:(5)$$SAR=\frac{\sigma |E^{2}|}{\rho }$$
where σ and ρ denote the electrical conductivity (S/m) and the mass density (kg/m^3^), respectively, while *E* is the electric field intensity (V/m). The SAR value is very important, as it depicts the effect of back radiation on human tissue. If it is more than the prescribed limit set by the FCC and ICINPR, then it will damage the tissue of the human body. At 2.45 GHz, when the antenna SAR value is simulated, it comes out as 0.19 W/kg, and at 5.8 GHz, it comes out as 1.18 W/kg for 1g of tissue, which is still within the permissible limits for 1g of tissue (see Figure 14). Thus, with the input power of 0.5 W, the SAR values of our antenna are within the acceptable range for both bands.

## 4. Results and Discussion

The antenna was simulated for use with human tissues and tested on a human hand experimentally. We noticed in both of the cases that the antenna shows good performance. This indicates that the antenna is stable on the human hand, and achieves a low SAR because of the full ground plane behind the substrate. The properties of the human tissues are listed in Table 3. The measured peak gain and the total radiation efficiency graph are given in Figure 15. From the graph, it can be observed that the gain of the antenna is more than 3 dBi at 2.45 GHz, while it is more than 6 dBi at 5.8 GHz, and the total radiation efficiency is more than 80% in both of the cases. A comparison of our proposed work with the previous related research is given in Table 4. If we compare the results of our proposed design with the previous works, it can be concluded that our design is more compact in size than other designs, and it also has a high gain. Additionally, the specific absorption rate (SAR) values are low, with 0.19 W/kg at 2.45 GHz and 1.18 W/kg at 5.8 GHz.

## 5. Conclusions

A compact dual-mode, dual-band patch antenna with L-shaped slots for on- and off-body communications in a wireless body area network is introduced in this paper. The radiation features of the proposed antenna have been introduced, and a simulation has been performed using computer simulation technology (CST). The proposed antenna is designed for on- and off-body communication links, and the material used for the substrate is FR-4, with a standard 1.6 mm thickness and a tangent loss of 0.02. The proposed antenna has been utilized to accomplish two diverse radiation modes at both frequencies. It has been noticed that at 2.45 GHz, the antenna’s radiation pattern is broadsided directional, whereas it is omni-directional at 5.8 GHz. The operational bandwidth of the presented antenna can reach up to 2.04% and 3.44% at 2.45 GHz and 5.8 GHz, correspondingly. The efficiency and performance of the designed antenna under the influence of a human body model were improved, and its SAR values were significantly reduced. Problems of compliance with safety related to on- and off-body communications are also discussed in brief detail. Hence, the proposed dual-band patch antenna is a good candidate for on- and off-body communication links.

## Figures and Tables

**Figure 1 sensors-21-07953-f001:**
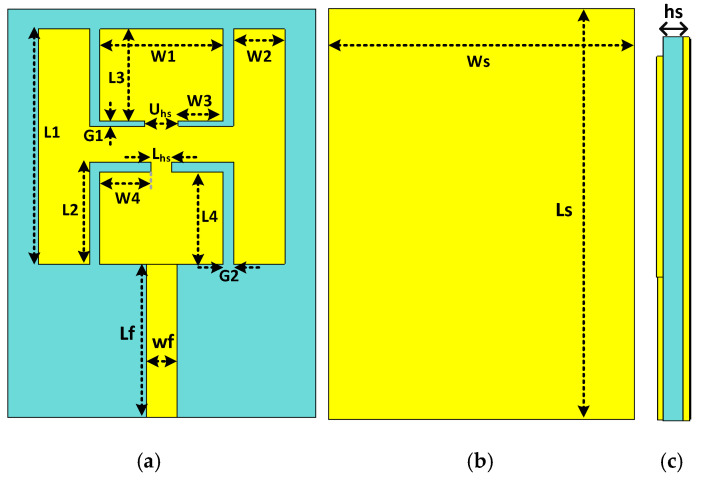
Antenna’s structure, (**a**) top layer, (**b**) back layer, (**c**) side view.

**Figure 2 sensors-21-07953-f002:**
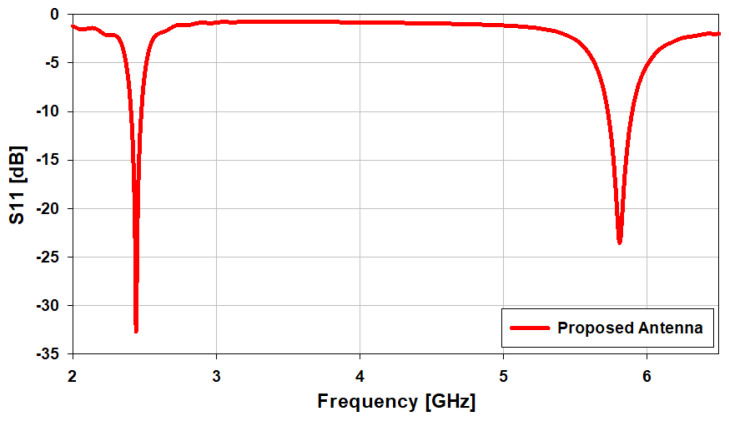
*S*_11_ of the designed dual-band antenna.

**Figure 3 sensors-21-07953-f003:**
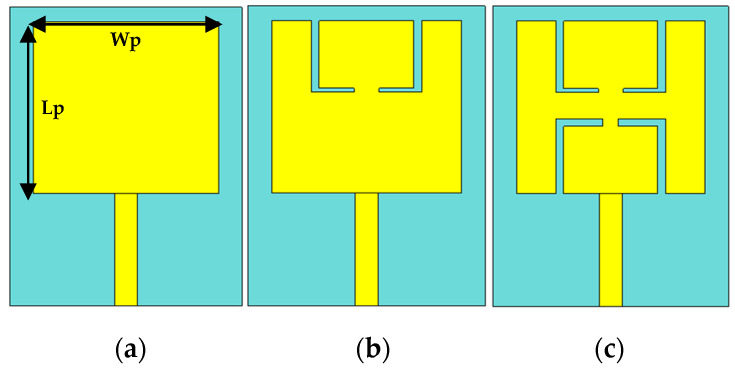
Proposed antenna design steps: (**a**) simple rectangular patch (ANT I), (**b**) patch with L-shape slots in the upper side (ANT II), (**c**) proposed patch (ANT III).

**Figure 4 sensors-21-07953-f004:**
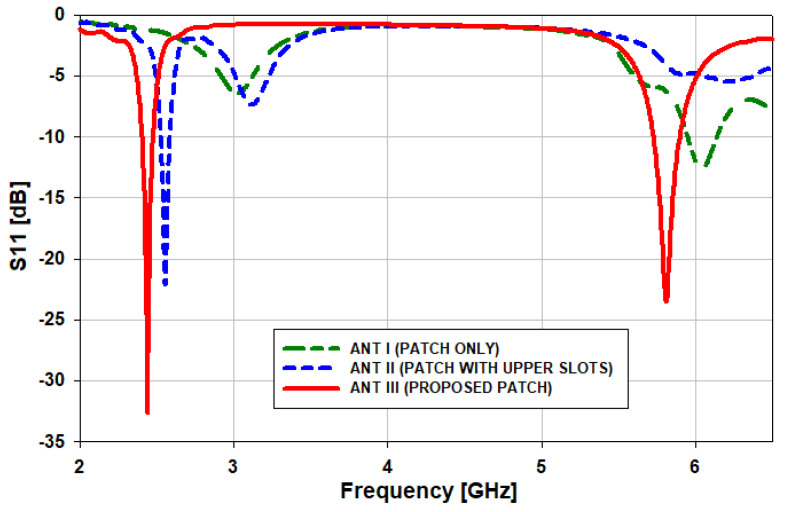
*S*_11_ comparison of the various antenna designs.

**Figure 5 sensors-21-07953-f005:**
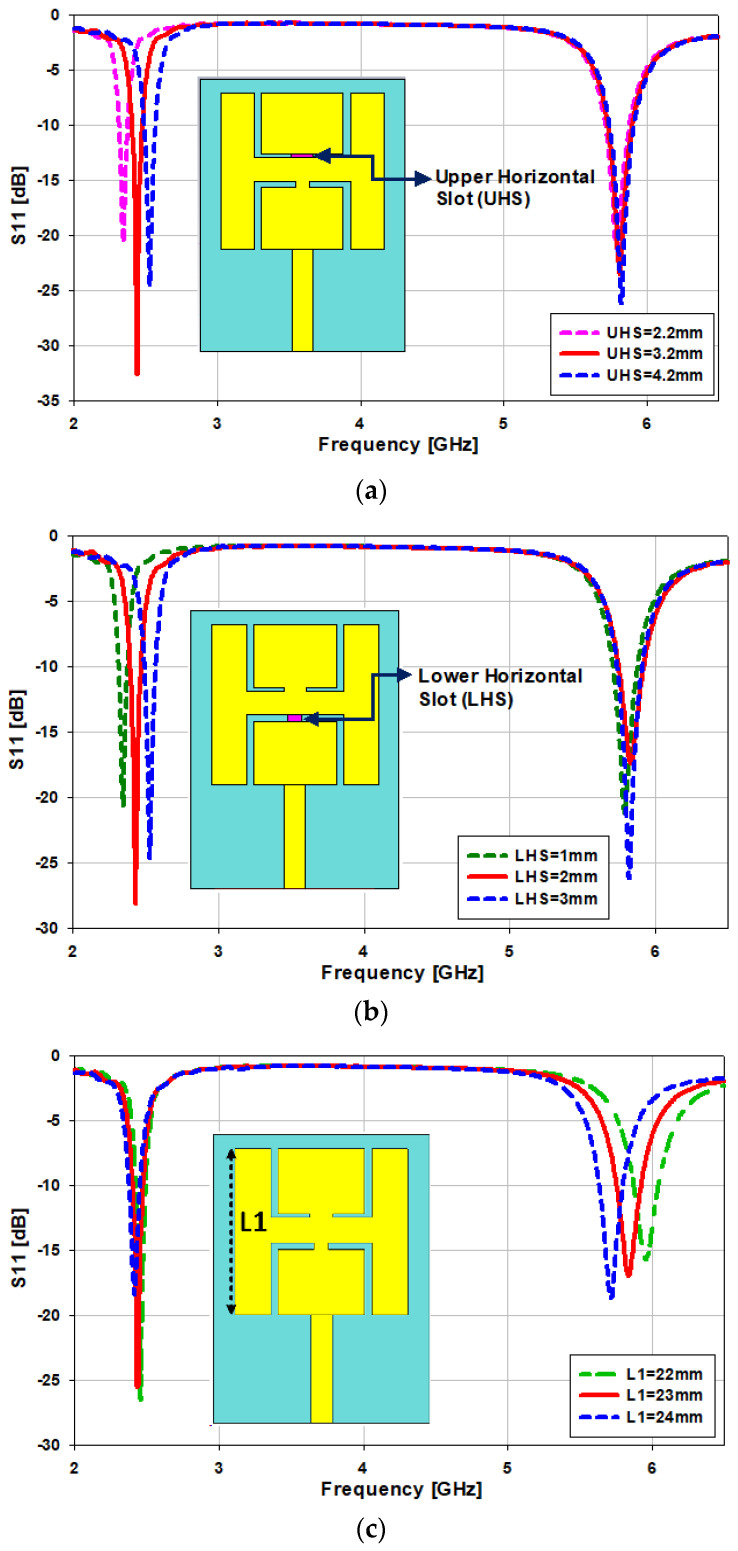
Optimization of the critical parameters of the antenna, (**a**) variation in the upper horizontal slot *“Uhs”*, (**b**) variation in the lower horizontal slot *“Lhs”*, (**c**) variation in the length of the patch *“L1”*, (**d**) variation in the width of the patch *“W1”*, (**e**) variation in the length of the slot *“L2”*.

**Figure 6 sensors-21-07953-f006:**
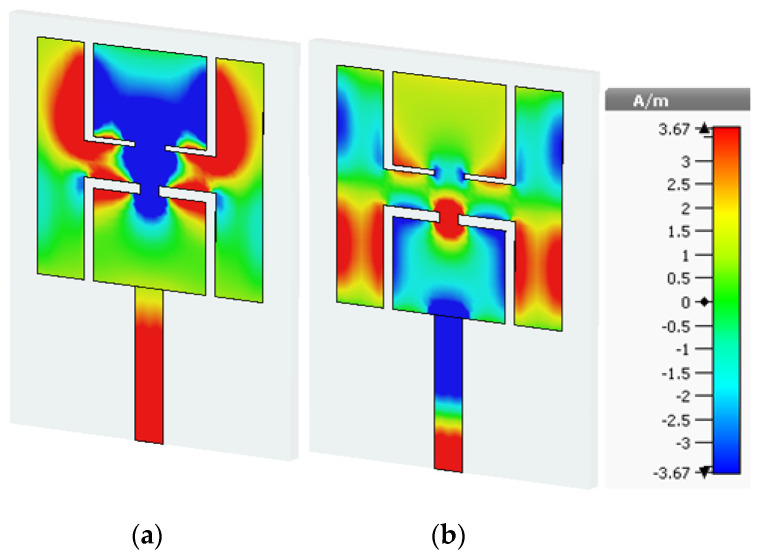
Surface current distribution (**a**) at 2.45 GHz and (**b**) at 5.8 GHz.

**Figure 7 sensors-21-07953-f007:**
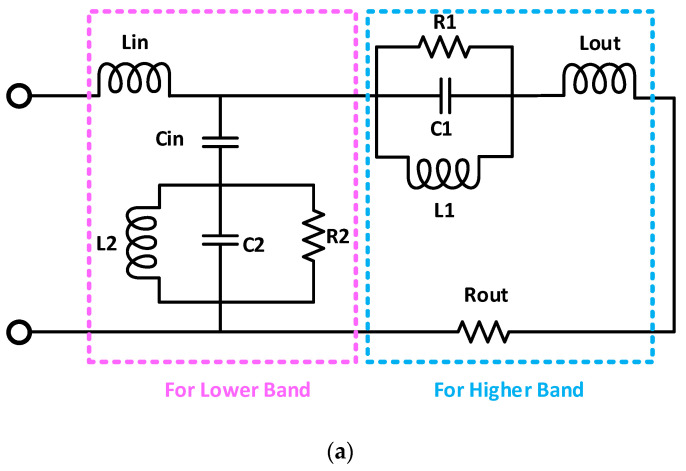
(**a**) Equivalent circuit model and (**b**) reflection coefficient of the equivalent circuit model.

**Figure 8 sensors-21-07953-f008:**
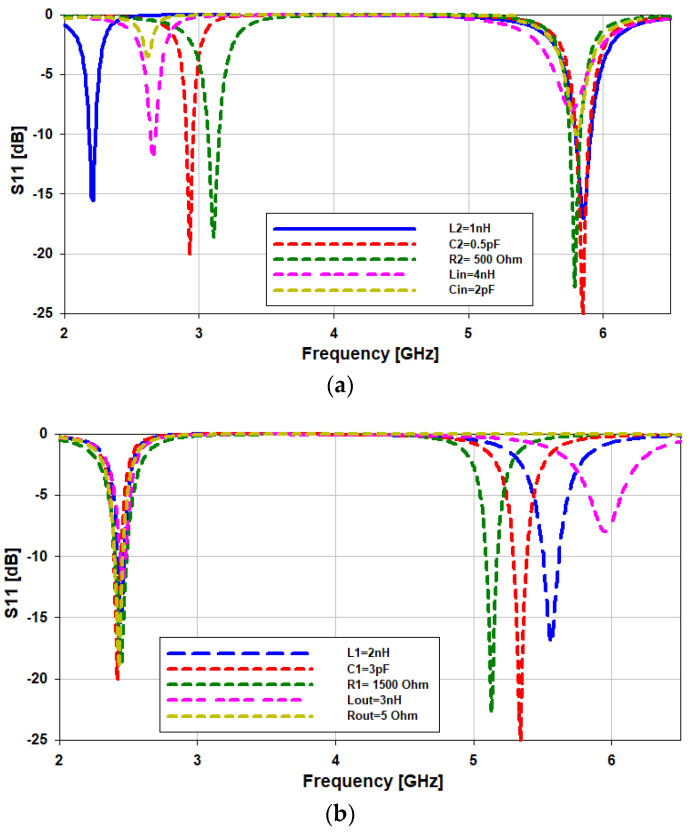
(**a**) Optimization of the circuit components for the lower frequency band and (**b**) the higher frequency band.

**Figure 9 sensors-21-07953-f009:**
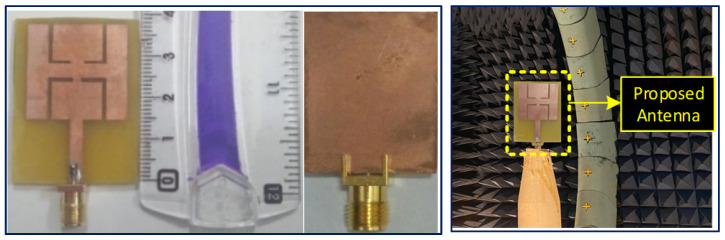
Fabricated prototype (front and back views) and the farfield evaluation setup inside the anechoic chamber.

**Figure 10 sensors-21-07953-f010:**
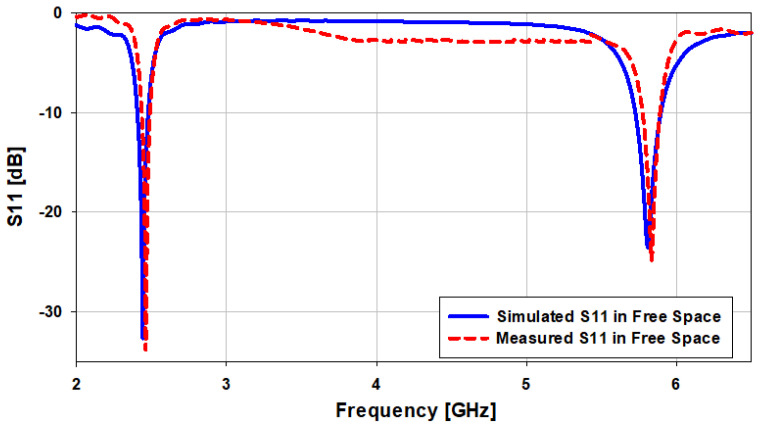
Comparison of the simulated and measured *S*_11_ in free space.

**Figure 11 sensors-21-07953-f011:**
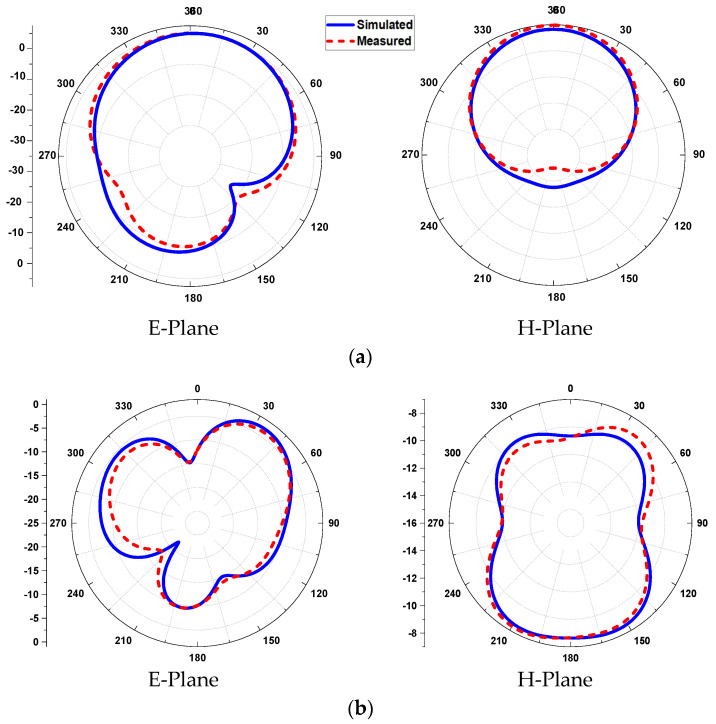
2D pattern of the antenna in free space, (**a**) at 2.45 GHz and (**b**) at 5.8 GHz.

**Figure 12 sensors-21-07953-f012:**
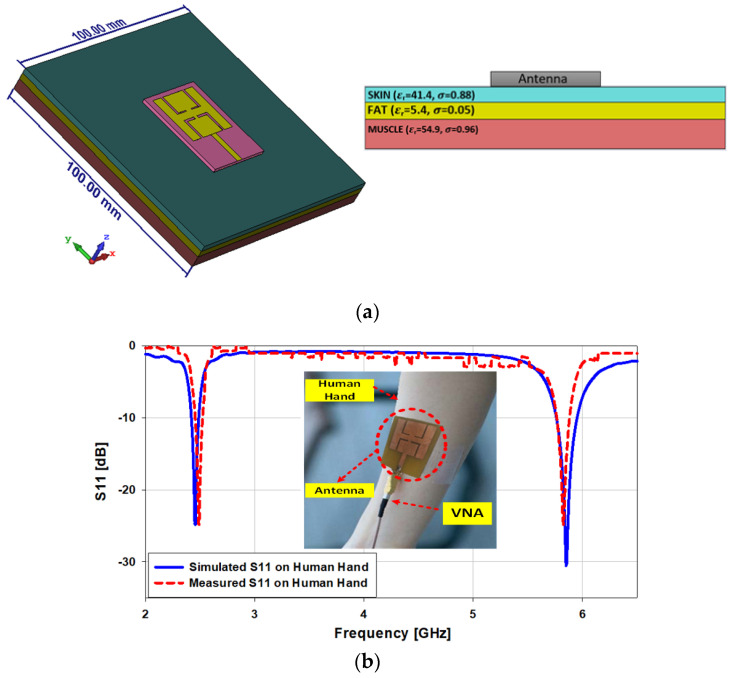
(**a**) Human body tissues; (**b**) comparison between the simulated and measured *S*_11_ on the human hand.

**Figure 13 sensors-21-07953-f013:**
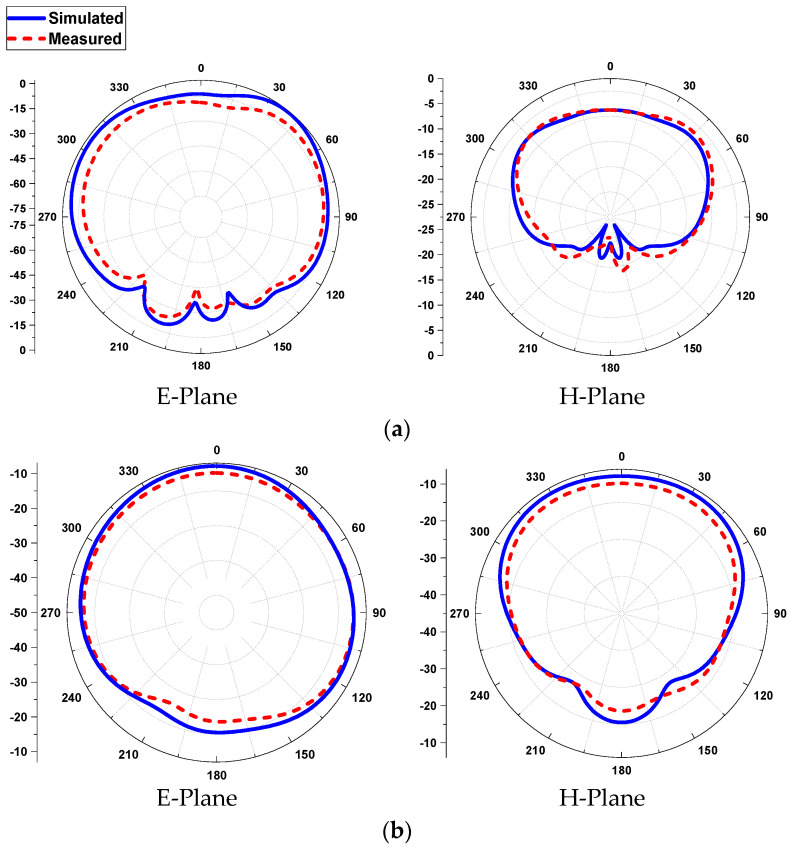
2D pattern of the antenna in human tissue, (**a**) at 2.45 GHz and (**b**) at 5.8 GHz.

**Figure 14 sensors-21-07953-f014:**
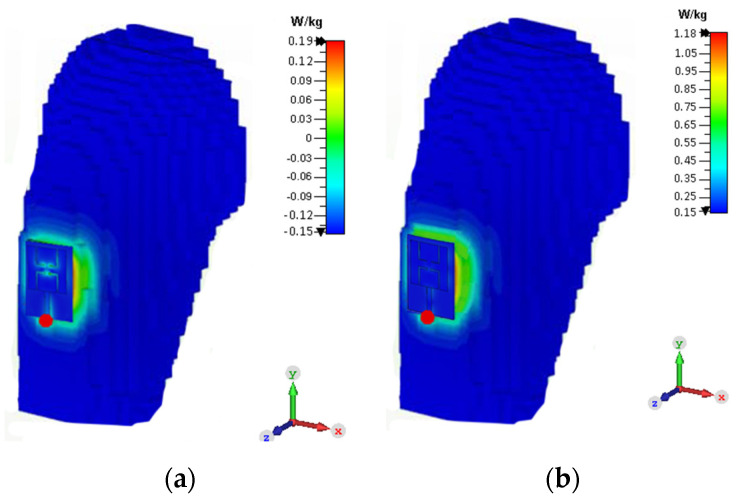
SAR distribution of the dual-band antenna; (**a**) at 2.45 GHz and (**b**) at 5.8 GHz.

**Figure 15 sensors-21-07953-f015:**
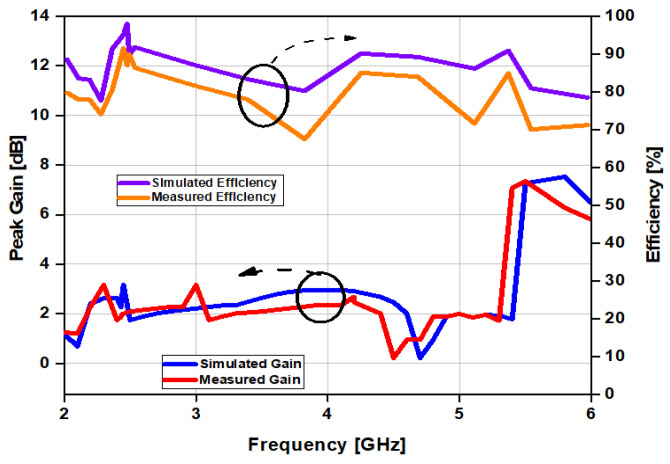
Simulated and measured gain and efficiency of the antenna on human tissues.

**Table 1 sensors-21-07953-t001:** Optimized parameters of the proposed antenna.

Parameters	Values (mm)	Parameters	Values (mm)
LS	40	Ws	30
Lf	15	Wf	3
L1	23	W1	12
L2	10	W2	5
L3	9	W3	4.4
L4	9	W4	5
g1	0.5	g2	1
Lhs	2	Uhs	3.2

**Table 2 sensors-21-07953-t002:** Values of the components used in the circuit model.

**Capacitors**	**Values (pF)**	**Inductors**	**Values (nH)**	**Resistors**	**Values (Ω)**
Cin	3	Lin	6	Zs	50
C1	2	L1	4.4	R1	2200
C2	1.115	L2	2	R2	760
		Lout	2	Rout	15

**Table 3 sensors-21-07953-t003:** Different human tissues’ properties [25].

Layers	Permittivity	Electrical Conductivity (S/m)	Density (kg/m^3^)	Thickness (mm)
Skin	41.3	0.895	1121	2
Fat	5.3	0.049	780	3
Muscle	54.8	0.955	1121	8

**Table 4 sensors-21-07953-t004:** Comparison of different on- and off-body antennas’ performances.

Ref. No.	Dimensions (mm^3^)	Frequency (GHz)	Substrate Material	Bandwidth (%)	Peaks Gain (dBi)	SAR (W/kg)	Proposed Technique
[17]	50 × 50 × 0.6	2.45/5.8	FR-4	4.2/ 10.5	1.2/7.9	0.81/0.24	Patch with Two Arms
[18]	100 × 100 × 2	2.45/5.8	Felt	11.9/2.18	6.33/6.98	0.042/0.09	Circular Patch
[19]	100 × 30 × 3.6	2.45/5.8	Leather	10.2/ 23.1	5.10/3.3	0.87/0.13	Belt-Shaped
[20]	30.5 × 62 × 3.15	2.45/5.8	Taconic TLY	3.47/2.58	1.51/6.44	--------	D-Shaped Patch
[21]	30 × 45 × 3.2	2.45/5.8	FR-4	4.9/2.8	3.09/0.64	--------	Parasitic Patches
[22]	80 × 92 × 2	2/5.8	Felt	9.48	8.26/9.86	--------	Triple Transmission Lines
[23]	100 × 100 × 3.2	2.45/5.8	F4B	2.57/5.22	1.9/5.9	0.254/0.074	Circular Patch
[24]	70 × 70 × 3	2.45/3.5	Felt	5.3/3.14	6	--------	Truncated Patch
[This work]	40 × 30 × 1.6	2.45/5.8	FR-4	2.04/3.44	5.08/6.33	0.19 /1.18	L-Slotted Patch

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
