# Peer review of "Compact Dual-Band Antenna with Paired L-Shape Slots for On- and Off-Body Wireless Communication"

_sensors, 2021, doi:10.3390/s21237953_

Round 1
Reviewer 1 Report
1. All abbreviations should be defined in the first place, e.g., SAR, ISM, WBAN. The same acronym in the abstract and another part of the paper should be defined in both abstract and another place in the first place.
2. The conclusion section is not quite interesting. It would be good to point out an interesting fact that would provide some inspiration to the reader.
3. The author should explain the diagrams of “Surface Current” at both bands i.e., 2.45GHz and 5.8GHz.
4. Authors should do the parametric study of the vertical slot used in the patch.
5. In table 3, do the authors used the same properties for both frequency bands, Please Explain!
6. The paper template should be further polished.
7. What is the purpose of the Equivalent Circuit Model?
8. In line 117 the authors said, “contains a 50-Ω CPW feedline, a rectangular patch, and the ground plane”, as there is no CPW technique utilized. Here you should mention it is a simple feedline.
Author Response
Dear Reviewer,
The authors are thankful for the Reviewer's detailed comments. It really helped us in the improvement of our manuscript. The authors have addressed all the comments of the Reviewer. The point-to-point response can be found in the attached file and the modifications have been highlighted in the revised manuscript.
Best Regards
Authors
Reviewer 2 Report
I want to thank the authors for the work presented. However, I see two weak points: 1) the publication of works similar to yours (refs 17 to 24) and 2) it would be advisable to perform a prototype that demonstrates that the designed antenna is suitable to perform On & Off-Body Wireless Communication, as suggested in the title of the work. In this way, the quality of the work would improve, making it possible to expand sections 4 and 5.
In addition, I have observed a series of possible errors that I have found in the document, which are the following:
- Line 86: the substrate is assumed to be between the patch and the ground plane, it is not necessary to specify it.
- Line 87: The electrical properties of FR4 are repeated again on line 101.
- Lines 17, 18, 89 and 90: (possible error or contradiction). Regarding radiation according to the band: in the low / high band, the authors say that it is broadband / omnidirectional (lines 17, 18). However, in lines 89 and 90 the authors say otherwise.
- Line 115: perhaps it goes without saying that the ideal value of S11 should be -infinite.
- Line 135: looking at the graph, the simple patch really resonates at 3 and 6 GHz. In any case, at 3 GHz it is not well matched as at 6 GHz.
- Line 210: focusing on the photograph of the antenna, is it the electrical contact of the SMA connector chassis with the ground plane guaranteed? In my opinion, it would be guaranteed if it was welded, although that does not mean that it works.
- Lines 235 and 294: the references relative to the electrical parameters of the tissues are missing: (skin, fat, muscle. See table 4).
- Line 271: define the SAR parameters and renumber the equation (5 instead of 1).
- Line 287: the author refers to the properties of the antenna listed in table 3. However, table 3 shows the electrical properties of the tissues.
Author Response

(The authors gave the same response as above.)

Reviewer 3 Report
The article is devoted to the design of dual band slotted microstrip patch antenna for Body Wireless Communication. The subject of design of such antennas is important and up-to-date, but has been exploited for years. Hundreds of scientific articles and dozens of books have been devoted to it. For this reason I have serious doubts whether the antenna described in the article can be treated as original enough. It is quite a classical microstrip design on cheap and high-loss FR4 laminate, which, for reasons unknown to me, the authors call 'biocompatible'. The authors do not cite the now classic literature on microstrip antennas, they only cite articles on body-centric communications or wearable antennas. In this way, the authors often cite secondary sources. For example, the entire theory regarding a classical patch antenna fed from a microstrip line (not CPW as it is written in the article!) can be found in C.A. Balanis, Antenna theory (...). Slotted patch antennas are discussed in the book Microstrip Patch Antennas of Lee Kai Fong, among others. Given the current state of the art, as the authors used typical methods and obtained fairly obvious results, I do not find the article of sufficient quality for publication in Sensors. Before possibly resubmitting the article, I suggest that the authors find answers to the following questions:
1) Why is this antenna biocompatible?
2) How does the described design differ from other known and described slotted patch microstrip antennas?
3) What is the purpose of building an equivalent circuit model if you don't give the relationships between the values of model elements and antenna dimensions?
4) Surely a large, relatively thick, rigid antenna is a good candidate for mounting on the human body?
Please also do a thorough literature review on similar designs.
Author Response

(The authors gave the same response as above.)

Round 2
Reviewer 2 Report
Thank you for reviewing your document. I only have one comment related to the reference to the permittivity of human tissues. I am not able to find this parameter in the references that you have indicated in your work, on the other hand, interesting works.
I think it would be appropriate to include a specific reference. I suggest one:
- Gabriel, R. W. Lau, and C. Gabriel, “The dielectric properties of biological tissues: II. Measurements in the frequency range 10 Hz to 20 GHz,” Phys. Med. Biol., vol. 41, no. 11, p. 2251, 1996
Author Response
Comment #1: Thank you for reviewing your document. I only have one comment related to the reference to the permittivity of human tissues. I am not able to find this parameter in the references that you have indicated in your work, on the other hand, interesting works. I think it would be appropriate to include a specific reference. I suggest one: Gabriel, R. W. Lau, and C. Gabriel, “The dielectric properties of biological tissues: II. Measurements in the frequency range 10 Hz to 20 GHz,” Phys. Med. Biol., vol. 41, no. 11, p. 2251, 1996.
Authors Response: We are thankful to the Reviewer for reviewing our manuscript. To address the Reviewer’s comment, we have added this reference as [23] in the updated manuscript.
Reviewer 3 Report
In my opinion, the article looks much better in its current form. However, I still have serious doubts about the equivalent circuit model. I am not convinced by the authors' explanation that 'The circuit model is designed to validate the antenna's scattering parameters and prove that our design is mathematically approved.' The agreement between the model and simulation results does not validate the results. The circuit model, being simplified, should explain the antenna properties in a simpler way. It may be helpful to include the relationships between the values of model elements and antenna dimensions. I expect such analyses from the authors in the final version of the paper. Furthermore, it seems to me that the rather poor agreement between the model and full-wave simulation results between the antenna operating bands can be improved by adding a parallel resistor to model the significant losses in the FR4 laminate.

Author Response
The authors are thankful to the Reviewer for the detailed review. We have updated the manuscript based on the Reviewer's comments. The detailed point-to-point to response to the Reviewer has been attached as a PDF file. However, the main modifications are listed below.
(1) The unnecessary sentence “The circuit model is designed to validate the antenna’s scattering parameters and prove that our design is mathematically approved” has been deleted to avoid any confusion.
(2) The antenna’s equivalent circuit diagram has been improved by mentioning the circuit for the lower band and upper band separately.
(3) A new section has been added to the revised manuscript to establish the relationship between the physical dimensions (key parameters) of the antenna with the RLC components in the equivalent circuit diagram.

Round 3
Reviewer 3 Report
Thanks to the authors for making the suggested corrections. In my opinion the manuscript can be accepted in the current version.